# Experimental Study on Texture Coupling Mechanism and Antifriction Performance of Piston Rod Seal Pair

**DOI:** 10.3390/mi13050722

**Published:** 2022-04-30

**Authors:** Jie Tang, Jie Zeng, Xin Lu

**Affiliations:** School of Aeronautical Engineering, Civil Aviation University of China, Tianjin 300300, China; jtang@cauc.edu.cn

**Keywords:** surface texture, high carbon chromium bearing steel (GCr15), ethylene propylene diene monomer (EPDM), friction, wear morphology, coupling

## Abstract

The effect of the coupling texture on the friction and wear of a piston rod-rubber seal pair under lubricating conditions is studied in this paper. Crescentiform textures with different area densities were fabricated on high carbon chromium bearing steel (GCr15) and ethylene propylene diene monomer (EPDM) materials by using a laser marking machine. We compare and analyze the effects of untextured, single-textured, and coupling-textured surfaces on the friction characteristics of the piston rod-rubber seal pair by conducting tests on the reciprocating module of the UMT-2 friction and wear testing machine. The results showed that the coupling-textured surface had the lowest coefficient of friction and wear compared to the untextured and single-textured surfaces. When the normal load was 10 N under the optimal coupling texture area density (6.4%), the friction and wear of the sealing pair decreased the most. Compared with the untextured surface, the friction coefficient was reduced by 27.9% and the wear amount was reduced by 30.0%; compared with the single-textured surface, the friction coefficient was reduced by 18.9%, and the wear amount was reduced by 23.8%. The coupling effect generated by the coupling texture effectively enhanced the formation and stabilization of the oil lubricant film and effectively captured wear debris, preventing it from continuously scratching the surface and reducing wear and roughness.

## 1. Introduction

A hydraulic cylinder is the executive component of an aircraft hydraulic system, and its internal piston rod sealing pair is a metal-rubber sealing pair, which is prone to wear failure under the combined action of hydraulic oil pressure, temperature, and viscosity, resulting in leakage; this affects the servo of the main rudder surface of the aircraft and, thus, the reliability and safety of the hydraulic system [1]. Therefore, the problem of how to further improve the lubricating properties of the metal–rubber sealing pair to reduce the wear of the piston rod needs to be solved urgently.

As a technology that can effectively improve the surface friction performance of oil-lubricated friction pairs, surface texture can effectively improve the wear of piston rod seal pairs [2,3]. Nowadays, there are various ways to process micro-texture structures. According to different mechanism categories, there are two main categories of ways to process micro-texture structures: one is the micro-texture processing technique based on material removal, and the other is the dimple texture processing technology that plastically deforms the material. The techniques of micro-texture processing based on material removal include electrolytic processing techniques [4,5,6,7], micro-electro-discharge machining [8,9,10,11], abrasive air-jet technology [12,13,14], and laser surface micromachining technology [15,16,17]. Plastic deformation technology of materials refers to the processing of micro-texture structures by applying a load to the base material that exceeds the range of elastic deformation under the action of an external force. Such processing techniques include conventional vibration-impact processing techniques [18,19,20], ultrasonic vibration machining techniques [21,22], embossing techniques [23], and laser shocking techniques [24,25,26,27,28]. Currently, laser surface micromachining techniques and electrolytic processing techniques are the mainstream texture processing techniques.

The mechanisms by which surface texture improves the tribological properties of materials include three main aspects: the ability to store lubricant under fluid lubrication conditions, improving lubrication conditions; the generation of hydrodynamic effect, increasing load-bearing capacity; and the storage of fine abrasive dust, reducing abrasive dust scratching. The morphology of the texture, geometrical parameters, and specific working conditions are the main parameters that influence the surface tribological properties [29]. Liao et al. [30,31] analyzed the influence mechanism of surface texture size parameters on the tribological properties of metal–rubber seal pairs by combining simulation and tribological experiments. The results showed that under the same conditions, the larger the working pressure and the larger the texture diameter, the greater the elastic deformation of the nitrile rubber and the better the tribological properties when the texture diameter was less than 300 μm. Wang’s team [32,33,34] showed that a surface texture with reasonable size parameters and arrangement could effectively reduce the friction and wear of metal–rubber sealing pairs; some surface texture samples exhibited better parameter combinations, where the friction coefficient, temperature increase, and wear could be reduced by more than 30%. Jiang et al. [35] used a ball-disk friction tester to study the lubrication characteristics of micro-pit PDMS friction pairs made by photolithography-complex mold technology. The results showed that under low-speed conditions, a smaller-diameter texture in the mixed-lubrication region could reduce friction, and the larger the area ratio of the pits in the experimental range, the more obvious the effect. He et al. [36] analyzed the tribological properties of textured metal and rubber specimens through experiments and showed that a reasonable texture combination and distribution could effectively improve the tribological properties of metal–rubber sealing pairs. Li et al. [37] designed a crescentiform surface texture and compared the optimization of water film carrying capacity and friction-reducing property for different rotational speeds and loads and different texture sizes by simulation. The results showed that the crescentiform surface texture of model CC1006 had the best-integrated optimization of water film carrying capacity and friction reduction performance. Tang et al. [38] redesigned the crescentiform surface texture and investigated the tribological properties of ethylene propylene diene monomer (EPDM) rubber under different loading conditions, and the results showed that under low loading conditions, the crescentiform surface texture can effectively reduce the friction coefficient of the friction pair and reduce corrosive wear and adhesive wear. Miao et al. [39,40] compared the friction performance of cylinder liner and piston ring (metal–metal seal) with single texture and coupling texture. Compared with single texture, the friction performance of coupling texture is better; the surface of the texture is easier to generate a stable oil film, which can enhance the ability to collect abrasive chips to prevent abrasive chips from scratching the surface and play a role in reducing friction.

At this stage, only the coupling effect of simultaneous texture in metal–metal seals has been studied, However, most studies on the texture of metal–rubber seals have focused on both as separate objects, exploring the effect of texture on its overall surface properties, respectively. The research involving the coupling effect of the simultaneous texturing of the two is relatively lacking. Therefore, this paper uses the reciprocating module of a UMT-2 friction and wear testing machine to compare the friction performances of the coupling-textured, untextured, and single-textured surfaces of a piston rod–rubber seal pair and analyze the coupling mechanism.

## 2. Experiments

### 2.1. Specimen Design

Figure 1 shows a schematic diagram of the piston rod sealing system. Under the action of hydraulic pressure, the piston rod and the rubber ring undergo reciprocating relative sliding, thereby converting hydraulic energy into mechanical energy. In order to simulate the reciprocating motion and working parameters of the piston rod and the rubber ring, the reciprocating module of a UMT-2 friction and wear testing machine was used. The upper sample was a pin that was made of high carbon chromium-bearing steel (GCr15), with a diameter of 3.6 mm and a height of 20 mm; the average surface roughness Ra measured after polishing is 0.2 μm. The lower sample was a disc made of EPDM material with a diameter of 24 mm, a thickness of 8 mm, and an elastic modulus of 7.8 MPa.

The crescentiform texture possesses good tribological properties with the advantages of both groove-shaped texture and discrete pit texture (refer to [36,41]). Figure 2a shows the 3D morphology of the crescentiform texture prepared by Cui et al. [41]. It can be seen that the crescentiform texture is more complex in shape, more difficult to process, and more costly for practical applications. Therefore, Tang et al. [38] simplified the crescentiform texture; the newly designed crescentiform texture consists of two concentric arcs, as shown in Figure 2b. The three-dimensional map and size map of the crescentiform texture in this paper are shown in Figure 2c,d, with an outer diameter of 200 μm and an inner diameter of 120 μm. The opening angle of the crescentiform pattern is *β* = 60°, and the area is *S_t_* = 0.1675 mm^2^ [38]. In order to obtain coupling textures with different domain densities, the texture spacing was changed while keeping the texture size constant. The area density of the texture is defined as the ratio of the area covered by the crescentiform texture to the area covered by the dummy cells, which is calculated as follows:(1)$$\alpha =\frac{S_{t}}{S}=\frac{0.1675}{l^{2}}$$
where α is the area density of the texture, $S_{t}$ is the area covered by the texture, S is the area covered by the cell, and l is the interval between the textures. To investigate the effect of coupling textures, texture patterns of different arrays were prepared by varying the texture area density of the GCr15 metal and EPDM rubber. The detailed texture pattern array parameters are shown in Table 1.

The coupling texture is a synergistic texture produced by the simultaneous texture of the two materials of the sealing pair. Therefore, a laser marker with a power of 75 W, a pulse frequency of 20 kHz, and a pulse width of 100 ns were used to process a crescentiform surface texture on the end face of the GCr15 pin; and a 50-W laser marker was used to process a crescentiform texture on the end face of the EPDM rubber. We washed and dried the test specimen with alcohol before texture processing. After processing, the texture heights at different positions were measured by a non-contact three-dimensional microscope, and the average value was calculated. Figure 3 shows the three-dimensional topography of the texture distribution on the surfaces of the GCr15 pin and the EPDM rubber.

### 2.2. Test Content

All tests were performed using the reciprocating module of a UMT-2 friction and wear testing machine, as shown in Figure 4. Before testing, the GCr15 metal material was lightly polished, and the GCr15 metal pins and EPDM rubber surfaces were cleaned with alcohol to remove dirt particles and then dried. The lower sample was fixed, and the upper sample moved back and forth via a motor. Test loads ranged from 10 to 40 N in 10 N increments. Since the normal movement speed of a hydraulic cylinder with rubber seals is 0.1–0.5 m/s [42], a movement speed of 0.4 m/s was selected. The movement amplitude was 10 mm, the frequency was 20 Hz, and the reciprocating friction test was carried out for 40 min [38]. All tests were carried out under lubricated conditions at room temperature (30 ± 2 °C). In order to provide sufficient lubricating conditions at the piston rod sealing interface, Castrol Magnetic Protection 0w-20 Oil was used; the lubricating oil was continuously supplied at a time interval of 5 s (12 drops per min) at the inlet area of the piston rod sealing contact. The normal load is set to 10, 20, 30, and 40 N (corresponding contact pressure is 1, 2, 3, and 4 MPa) [30]; the detailed test conditions are shown in Table 2.

The normal load and friction force were measured by the *Z*-axis and *XY*-axis force sensors, respectively. After each test, the friction coefficient and wear amount of the EPDM sample were calculated using Equations (2) and (3). Each test was carried out three times, and the average values of the friction coefficient and wear amount were taken to ensure the accuracy of the test. The EPDM samples were sprayed with gold, and the sampling range was 0.8 × 0.8 mm. The wear scars and wear debris were observed by FE-SEM. The distribution of elements on the surface of the GCr15 metal after the test was measured by EDS.
(2)$$Friction\;coefficient=\frac{F_{f}}{F_{n}}$$
(3)$$Abrasion=W_{b}-W_{a}$$
where $F_{f}$ is the friction force, $F_{n}$ is the normal load, $W_{b}$ is the weight of the EPDM sample before the test, and $W_{a}$ is the weight of the EPDM sample after the test.

In order to verify the test method in this paper, single-textured specimens with a rubber surface texture area rate of 6.4% were selected for frictional wear tests under different contact pressure conditions, and the variation curve of friction coefficient with contact pressure was obtained and compared with the test data reported in the literature [38]; the comparison graph is shown in Figure 5. The “Single texture-Rubber” data in the figure is the test data of the textured sample only on the rubber. Under the same experimental conditions, the two sets of experimental data agreed well.

## 3. Results and Discussion

### 3.1. Effect of normal load

The effect of normal load on friction and wear behavior was investigated. Figure 6 shows the experimental results of the friction behavior of the piston rod seal specimens under different normal loads in the range of 10–40 N, a texture depth of 5 μm, and a sliding speed of 0.4 m/s. A textured area density of zero represents an untextured sample. As the normal load increased from 10 to 40 N, the friction coefficient of the coupling-textured sample increased gradually. This is because the larger the normal load is, the more the volume of rubber in the untextured area is pressed into the texture of the metal dimples. Then, the scraping and cutting effect is more obvious during the reciprocating motion of the rubber, and stress concentration occurs at the edge of the rubber texture, which leads to aggravated wear of the rubber texture edge, increases the surface roughness of the specimen, and then leads to an increase in the friction coefficient. The friction coefficient of the untextured specimen has a slowly decreasing trend, and the laser surface texture pattern in the literature [38] also has a similar trend. Compared with the untextured specimen, when the normal load was less than 20 N, the friction coefficient of the coupling-textured specimen decreased significantly, while under a normal load greater than 20 N, it increased significantly. The minimum coefficient of friction was obtained on a textured array with a normal load of 10 N and a coupling texture area density of 6.4%. At lower normal loads (i.e., less than 20 N), the friction coefficient of the coupling texture sample decreased by 27.9% compared to the untextured specimen. Figure 7 shows the variation of the wear amount of samples with coupling texture area densities under different normal loads. It can be seen that when the area density of the coupling texture was in the range of 6.4–25.6%, the wear amount and friction coefficient of the sample had a similar trend with the normal load, and the wear amount increased with increasing normal load. When the normal load is 40 N, the wear amount is the largest. This is due to the increased wear caused by stress concentration at the edge of the texture; a large amount of wear debris is generated, which exceeds the wear debris capture ability of the coupling texture, causing the large wear debris to continue to damage the surface. This, in turn, leads to increased wear. The wear amount of the untextured specimen increased gradually with increasing normal load. Among all coupling textured arrays, the specimens under a normal load of 10 N had the lowest wear. When the coupling texture area density was 6.4% and the normal load was 10 N, the wear amount decreased by as much as 30.0% compared with the untextured specimen.

### 3.2. Effect of Coupling Texture Area Density

The effect of different texture area densities on the friction and wear properties of the coupling-textured and single-textured samples under a normal load of 10 N was investigated by changing the texture spacing without changing the texture size or height. Figure 8 shows the variation curves of friction coefficient vs. the area density of the samples under a normal load of 10 N and a texture depth of 5 μm. In the figure, “Single texture-Metal” indicates that the texture was only on the GCr15 metal (i.e., the upper sample). The results show that the friction coefficient of the experimental group textured only on the metal increased with area density, and the friction coefficient was lowest when the areal density of the textured array was 6.4%. The friction coefficient of the test group textured only on the rubber decreased first and then increased with increasing area density. When the area density of the textured array was 12.8%, the friction coefficient decreased significantly. When the area density of the textured array was greater than 12.8%, the friction coefficient decreased. The friction coefficients were all smaller than those of the experimental group textured only on the metal, which is consistent with the research in the literature [43]. Under a normal load of 20 N, the friction coefficient of the coupling-textured specimens increased with area density, and the friction coefficient was the smallest when the area density of the textured array was 6.4%. Compared with the single-textured samples, the friction coefficients of the coupling-textured samples were significantly reduced, with a maximum reduction of 18.9%. It can be seen that the coupling texture plays a positive role in the reduction of the friction coefficient, which is due to the coupling texture that enhances the replenishment of the oil film on the contact surface; the dynamic pressure effect is more obvious so that the friction coefficient is reduced. However, the increase of the area density of the coupling texture will reduce the bearing area, increase the specific pressure, and increase the friction coefficient, so that the friction coefficient increases with the increase of the area density. Figure 9 shows the variation curve of the wear amount vs. the area density of the sample with a texture depth of 5 μm under a normal load of 10 N. The variation trends of the wear amount and friction coefficient of the sample vs. texture area density were similar. The wear amount of the coupling texture increased with the increase in the area density of the textured array. When the normal load was 10 N, the area density was 6.4%, and the structural array experienced the least amount of wear. Compared with the single-textured samples, the wear of the coupling-textured samples all decreased, with a maximum decrease of 23.8%. This showed that under certain conditions, the effect of a coupling texture on wear reduction was more significant than that of a single texture.

### 3.3. Analysis of Wear Morphology

Figure 10 shows the typical morphologies of the worn surfaces of the EPDM materials under a normal load of 10 N. Figure 10a shows the worn surface of the EPDM material under the untextured condition, where rubber-specific pattern wear marks appeared on the surface of the sample. The overall pattern was relatively flat and regular. The wear debris adhered during the friction process acts as a solid lubricating layer in the friction pair and is discharged from the friction area during the subsequent friction process. Thus, uniform friction is achieved. Figure 10b shows the worn surface of the EPDM material is only on the surface of the GCr15 metal, and the texture area density is 6.4%. A large number of small corrosion pits were distributed on the surface of the material, mainly because the hydraulic oil eroded the surface of the EPDM material during the friction process, which is a typical manifestation of corrosion wear. The surface of the material had obvious flake peeling, indicating that adhesive wear had occurred during the friction process. There was a large amount of wear debris on the surface of the EPDM material, and there was a large amount of wear debris. This was because the texture on the surface of the GCr15 metal only collected part of the wear debris, indicating that the ability of the single texture to collect and store wear debris was limited. Figure 10c shows the worn surface of the EPDM material under the condition that only the EPDM rubber surface was textured, with a texture area density of 6.4%. There was wear on the textured edge, as well as crescent-shaped protrusions in the vertical direction, indicating that adhesive wear mainly occurred during the grinding process. Furthermore, there was more wear debris on the textured edge and the untextured area. Thus, the wear debris was not completely stored in the texture, and larger debris could further damage the surface. Figure 10d shows the worn surface of the EPDM material under the coupling texture treatment, with a texture area density of 6.4%. It can be seen that there was no obvious wear on the textured surface, and only a small amount of wear scars and small corrosion pits appeared in the untextured area. The wear mechanism was mainly corrosion wear. The wear debris was mainly concentrated in the texture, and there was basically no wear debris in the untextured area. This was because the coupling effect improved the ability of the texture to collect and store the wear debris, which effectively prevented large wear debris from damaging the surface.

### 3.4. Morphology Analysis of Wear Debris

Wear debris can reflect the characteristics of the friction pair system and is an important information carrier. The analysis of wear debris can map the wear characteristics of a friction pair surface [44]. Figure 11a shows the wear debris morphology of the EPDM material of the untextured sample after wear under a normal load of 10 N. It can be seen that the main body of the wear debris was flake. It is generally believed that flake wear debris is the product of wear debris between friction pairs being rolled and ironed under the action of pressure, and its wear mechanism is mainly adhesive wear. Figure 11b shows the wear debris morphology of the EPDM material after wear under a coupling texture area density of 6.4% and a normal load of 10 N. Through observation, its overall size was slightly smaller than that of the other wear debris; it was peanut-shaped, and its surface was relatively smooth, which is typical of corrosion wear debris, where chemical reactions such as oxidative degradation and molecular bond breakage occur on the rubber surface. Under lower loads, the lubricating oil stored in the coupling texture could produce a good hydrodynamic pressure effect, and the oil film pressure and the normal load could be balanced, which could improve the formation and stability of the oil film. At the same time, the coupling texture had a good ability to collect and store wear debris, which avoided the continued damage from large wear debris, resulting in minimal surface wear. Figure 11c shows the wear debris morphology of the EPDM material after wear under a coupling texture area density of 32.0% and a normal load of 40 N. It can be seen that the wear debris under these conditions was strip-shaped, with obvious characteristics of cutting wear debris. Generally speaking, cutting wear particles are formed by the plowing of hard and sharp asperities on a softer wear surface [45]. Relevant studies [46] show that the introduction of surface texture will produce stress concentration at its edges, and when the normal load is large, significant bulges will occur, which may lead to the occurrence of micro-cutting. Here, the shape of the wear debris proved that under poor working conditions and coupling texture parameters, serious furrows and even cutting wear effects did indeed occur.

### 3.5. Energy Spectrum Analysis

Figure 12 presents electron microscope images and energy spectrum analyses of the GCr15 material under a normal load of 10 N for the untextured sample. Adhesive wear was indicated by dark flakes, and localized tears were featured along the sliding direction. From the energy spectra, it can be seen that the content of C and O elements on the surface of the GCr15 metal sample was relatively high, indicating that the polymer material had been transferred during the friction process. Figure 13 shows electron microscope images and energy spectrum analyses of the GCr15 material under a coupling texture area density of 6.4% and a normal load is 10 N. The wear surface around the dimples was smoother and less scratched. At this time, the content of C and O elements at the edge of the pit texture decreased compared with that of the untextured sample, indicating that the adhesion and transfer phenomenon of the polymer material was improved. Figure 14 shows electron microscope images and energy spectrum analyses of the GCr15 material under a coupling texture area density of 32.0% and a normal load of 40 N. There were many scratches on the surface of the sample, and serious wear occurred around the pit texture. The content of the C element at the edge of the pit texture was high, while that of O was low, indicating that during the grinding process, due to the stress concentration on the edge of the pit, the local high temperature made the EPDM material burn and degenerate, resulting in severe cutting wear, which was consistent with the results of the wear debris morphology analysis.

### 3.6. Mechanism of Texture Coupling

The schematic diagram of the coupling texture mechanism (at the symmetry plane) is shown in Figure 15. Before the GCr15 metal dimple texture reached the EPDM rubber dimple texture, the lubricating oil stored in the EPDM rubber dimple texture formed an oil film on the untextured area. With the relative sliding of the friction pair, the GCr15 metal dimples collected the wear debris generated by part of the wear, which led to the weakening of the dynamic pressure effect in the dimple texture and a decrease in oil pressure. When the GCr15 metal dimple texture passed through the EPDM rubber dimple texture, the normal load extruded the lubricating oil from the EPDM rubber dimple, filling in the GCr15 metal dimple texture and replenishing the oil film. At the same time, the inside of the GCr15 metal dimple was pushed out. Part of the collected wear debris was flushed into the underlying rubber dimple texture, preventing the wear debris from continuing to damage the surface as the metal pin moved. When the GCr15 metal dimple texture left the EPDM rubber dimple texture, due to the lubricating oil supplemented by the EPDM rubber dimple texture, the lubricating oil in the GCr15 metal dimple texture enhanced the dynamic pressure effect and increased the maximum oil film bearing capacity. The reduction of wear debris in the metal pit texture also relatively enhanced its ability to capture new wear debris. Under the coupling effect of the metal dimple texture and the rubber dimple texture, the dynamic pressure effect inside the coupling texture increased, which increased the maximum pressure that the oil film could bear, forming a virtuous circle and improving the formation of the oil film on the contact surface during the entire friction process. After the coupling texture treatment discussed in Section 3.3, the distribution area of the wear debris on the worn surface of the EPDM material under an area density of 6.4% proved the wear debris capture mechanism of the coupling texture.

## 4. Conclusions

This paper presents the effects of coupling textures on the friction and wear of the piston rod–rubber seal pair under different normal loads; different texture area densities are introduced. We used a laser marker to create crescentiform textures on the surfaces of the two materials of the sealing pair and performed a reciprocating sliding test. The effects of untextured, single textured, and coupling textured surfaces on the friction characteristics of the piston rod–rubber seal pair were compared and analyzed. This study draws the following conclusions:

(1) The reduction in friction coefficient is inversely proportional to the normal load. Under lubricating conditions, the friction coefficient increases linearly with the increase of the normal load; under low normal load conditions, the magnitude of the friction coefficient is proportional to the area density of the coupling texture. The wear on the surface increases linearly.

(2) Under the condition of low normal load, the coupling texture array has a significant reduction effect on the friction coefficient and wear amount compared with the untextured and single textured surfaces. When the area density of the coupling texture is 6.4% and the normal load is 10 N, the friction and wear of the sealing pair decrease the most. Compared with the untextured surface, the friction coefficient decreased by 27.9% and the wear amount decreased by 30.0%; compared with the single-textured surface, the friction coefficient decreased by 18.9% and the wear amount decreased by 23.8%.

(3) The coupling effect generated by the coupling texture can effectively enhance the formation and stabilization of the oil film, produce a good dynamic pressure effect, and has a good ability to capture wear debris, prevent the wear debris from continuously scratching the surface, and reduce wear and roughness. The service life of the piston rod seal pair is improved.

## Figures and Tables

**Figure 1 micromachines-13-00722-f001:**
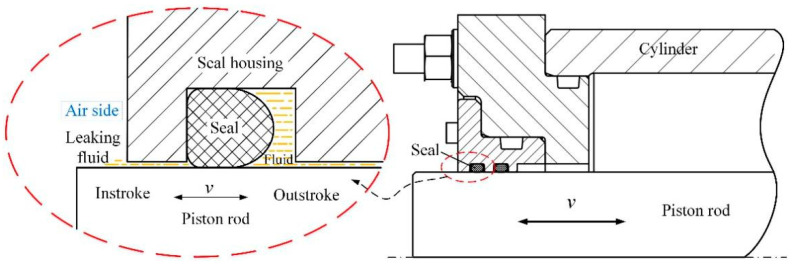
Schematic diagram of the piston rod sealing system.

**Figure 2 micromachines-13-00722-f002:**
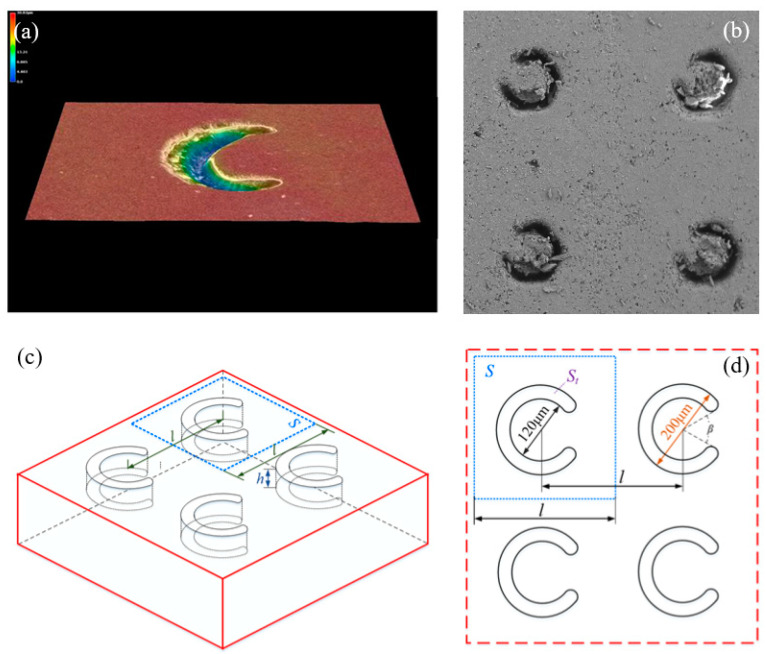
Schematic diagram of the crescentiform surface: (**a**) the crescentiform surface prepared by Cui et al., (**b**) the crescentiform surface prepared by Tang et al., (**c**) the three-dimensional schematic diagram of the crescentiform surface in this experiment, and (**d**) the crescentiform surface dimensions.

**Figure 3 micromachines-13-00722-f003:**
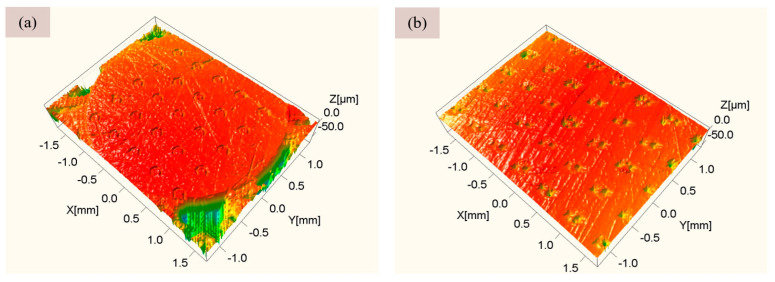
Three-dimensional topography of surface textures of the carbon chromium bearing steel (GCr15) pin and ethylene propylene diene monomer (EPDM) rubber: (**a**) GCr15 pin surface with a texture area density of 6.4%; (**b**) EPDM rubber surface with a texture area density of 6.4%.

**Figure 4 micromachines-13-00722-f004:**
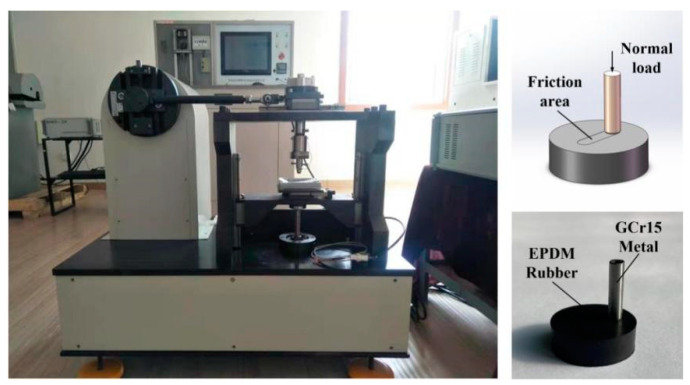
UMT-2 reciprocating test device.

**Figure 5 micromachines-13-00722-f005:**
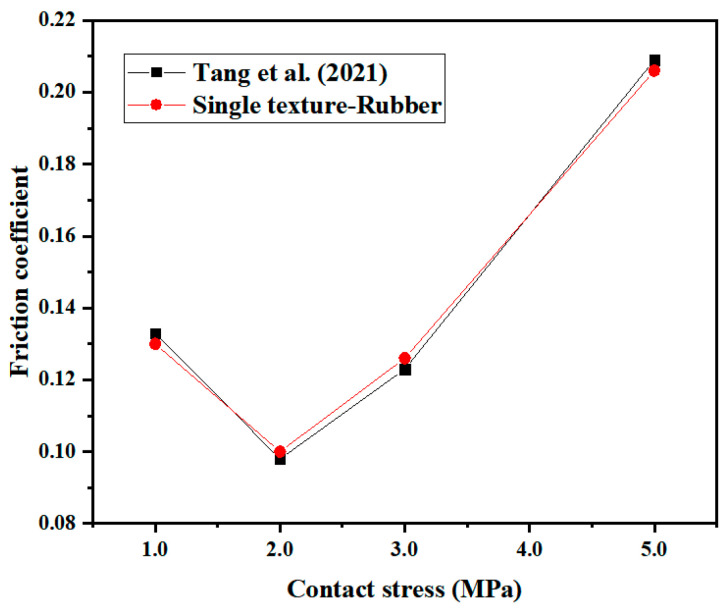
Comparison of the influence of the crescentiform texture on the friction coefficient under different contact pressures.

**Figure 6 micromachines-13-00722-f006:**
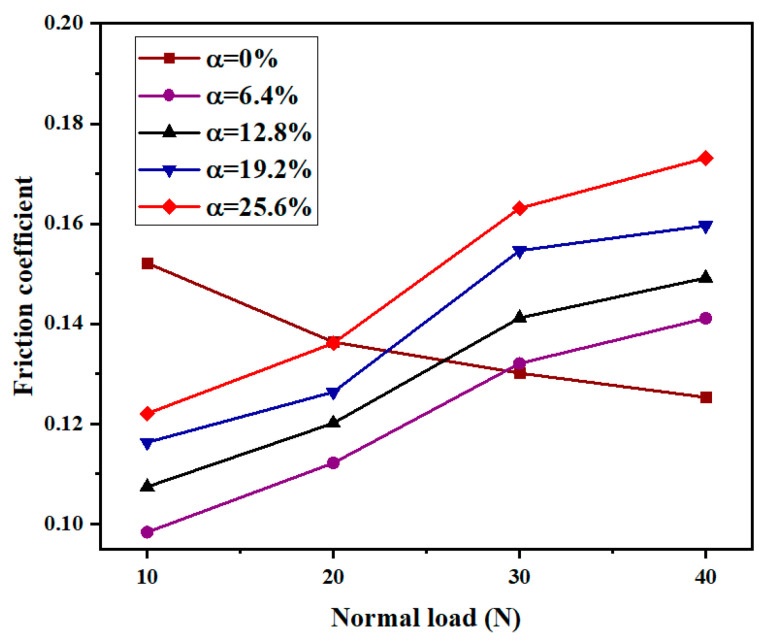
Variation of friction coefficient with normal load.

**Figure 7 micromachines-13-00722-f007:**
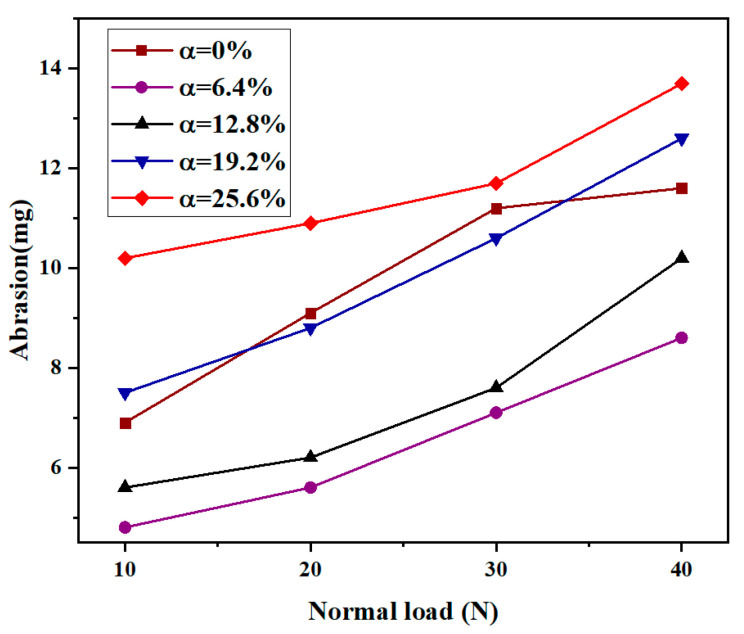
Variation of abrasion with normal load.

**Figure 8 micromachines-13-00722-f008:**
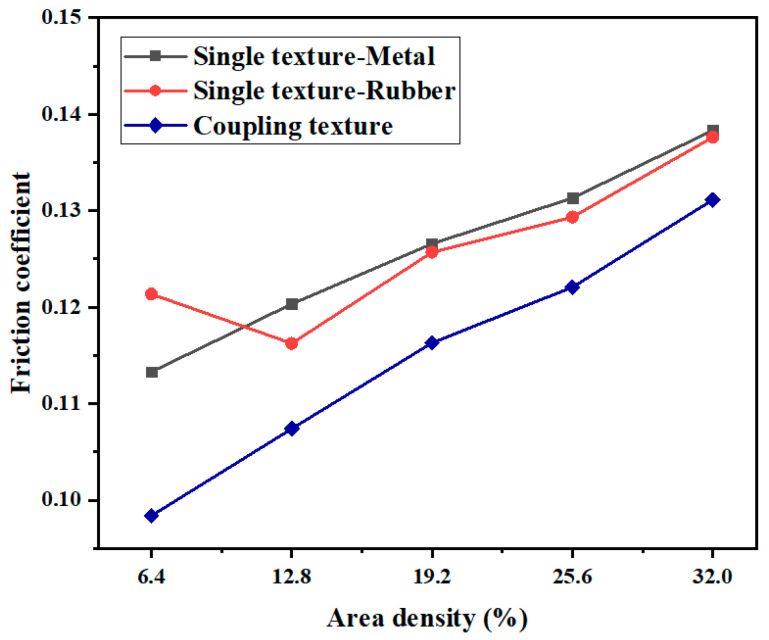
Variation of friction coefficient with area density.

**Figure 9 micromachines-13-00722-f009:**
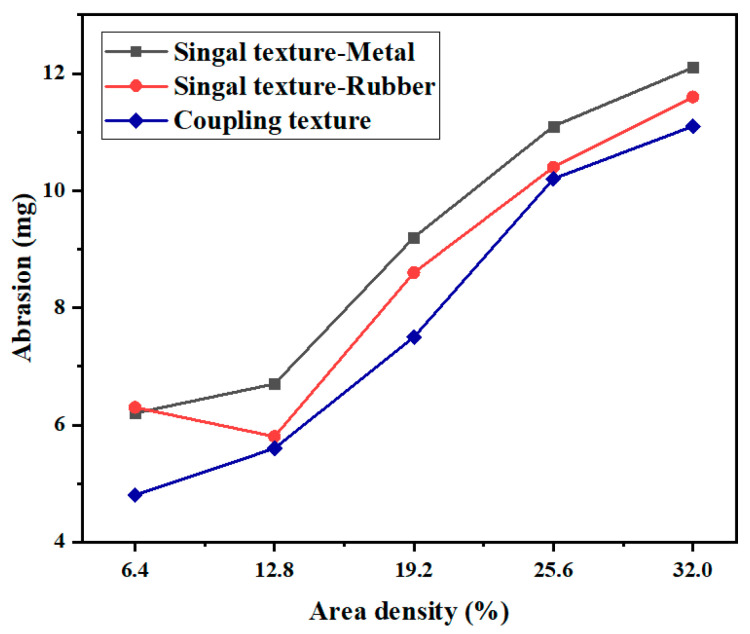
Variation of abrasion with area density.

**Figure 10 micromachines-13-00722-f010:**
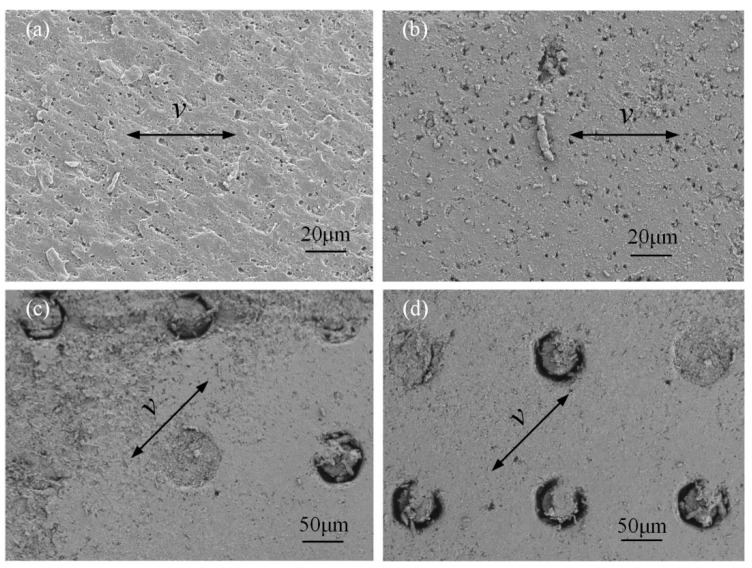
Surface topography of EPDM material after abrasion under conditions of (**a**) untextured, (**b**) single texture on GCr15 metal, (**c**) single texture on EPDM rubber, and (**d**) coupling texture parameters.

**Figure 11 micromachines-13-00722-f011:**
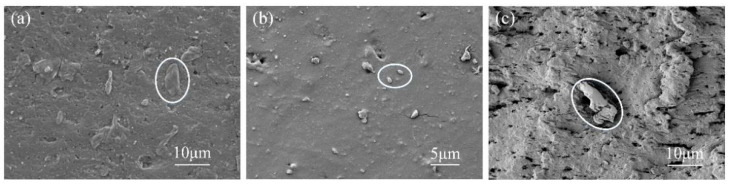
SEM images of wear debris of EPDM material: (**a**) morphology of wear debris of untextured samples; (**b**) wear debris topography with optimal coupling texture parameters; (**c**) wear debris topography with worst coupling texture parameters.

**Figure 12 micromachines-13-00722-f012:**
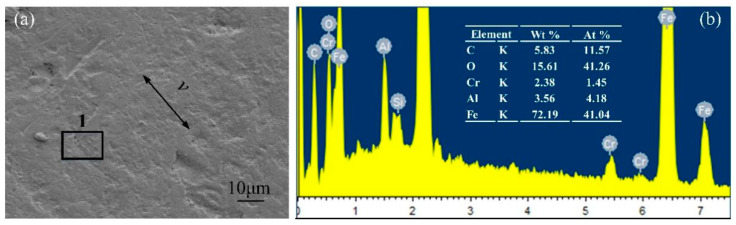
SEM images and EDS analysis of GCr15 metal surface after wear of untextured samples: (**a**) SEM image of untextured sample surface; (**b**) corresponding EDS analysis.

**Figure 13 micromachines-13-00722-f013:**
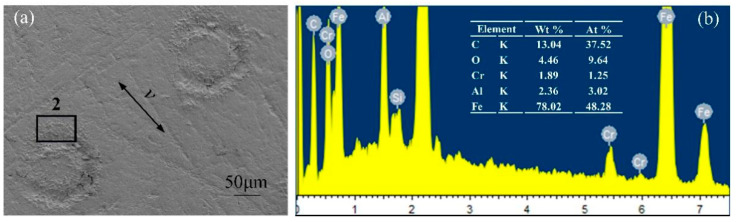
SEM images and EDS analysis of GCr15 metal surface under optimal coupling texture parameters: (**a**) SEM image of the sample surface with optimal coupling texture parameters; (**b**) corresponding EDS analysis.

**Figure 14 micromachines-13-00722-f014:**
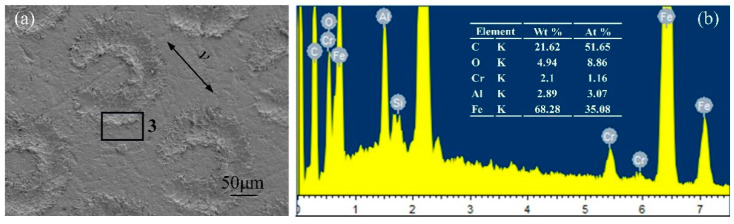
SEM images and EDS analysis of GCr15 metal surface under the worst coupling texture parameters: (**a**) SEM image of the sample surface with the worst coupling texture parameters; (**b**) corresponding EDS analysis.

**Figure 15 micromachines-13-00722-f015:**
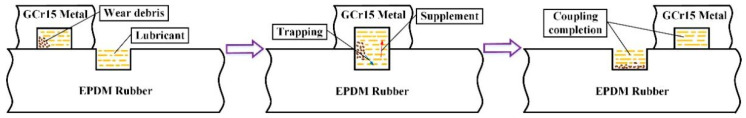
Action mechanism of coupling texture.

**Table 1 micromachines-13-00722-t001:** Parameters for the array of textured pattern.

Parameters	**Coupling Texture Area Density *α***	Texture Height *h*
for all samples	6.4%, 12.8%, 19.2%, 25.6%, 32.0%	5 μm

**Table 2 micromachines-13-00722-t002:** Friction test conditions.

Specification	Value
Normal load	10 N, 20 N, 30 N and 40 N
Speed	0.40 m/s
Lubricant	Castrol Magnetic Protection 0w-20 Oil
Displacement amplitude	10 mm
Room temperature	30 ± 2 °C

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
