# Peer review of "Experimental Study on Texture Coupling Mechanism and Antifriction Performance of Piston Rod Seal Pair"

_micromachines, 2022, doi:10.3390/mi13050722_

Round 1

Reviewer 1 Report

Reviewer’s comments

Journal- Micromachines

 Manuscript Number- 1678506

Title of Paper-   ‘Experimental study on texture coupling mechanism and antifriction performance of piston rod seal pair”

 The paper concerns with experimental investigation related to the effect of different types of texture patterns on the friction and wear of a piston rod-rubber seal pair. The experiments were carried out using the reciprocating module of a UMT-2 friction and wear testing machine

Comments to the Author:

After going through the manuscript, the reviewer comes up with the following observation.

  • More recent and relevant studies need to be discussed in the introduction section.
  • How the selections of the operating and geometric parameters have been carried out in the study? Do these parameters follow some standards?
  • Why the validation of the present experimental results with earlier published articles is not presented? How the author clarifies the validity of work? There is no information, on the authenticity and validity of the adopted experimental methodology in the present work?
  • Most of the description in this paper was spent for describing the trends obtained from experimental results. The author(s) should compute the experimental results with physical explain for give more strength in paper.
  • Conclusions of the manuscripts are not up to the mark. Conclusions need to be revised. So as to make them more meaningful.

In the opinion of the reviewer, the author needs to suitably incorporate all these major and mandatory changes in the revised manuscript.

Reviewer 2 Report

The results are interesting, but the article requires some clarifications:

  • How the friction factor Ff is measured?
  • I suggest converting the load (N) shown in Figures 4 and 5 into unit pressures (MPa). It would be easier to compare the results of this work with the results obtained by the other researchers
  • In Figures 6 and 7 it would be worth to add the results for the untextured samples.
  • How long did one test last? and are these results sufficient to assess the durability/wear of the tested material solutions?
